# Contribution of Hydrogen Cyanide to the Antagonistic Activity of *Pseudomonas* Strains Against *Phytophthora infestans*


**DOI:** 10.3390/microorganisms8081144

**Published:** 2020-07-28

**Authors:** Abhishek Anand, Delphine Chinchilla, Christopher Tan, Laurent Mène-Saffrané, Floriane L’Haridon, Laure Weisskopf

**Affiliations:** Department of Biology, University of Fribourg, 1700 Fribourg, Switzerland; abhishek.anand@unifr.ch (A.A.); delphine.chinchilla@unibas.ch (D.C.); christopher.tan@unifr.ch (C.T.); laurent.mene-saffrane@unifr.ch (L.M.-S.); floriane.lharidon@unifr.ch (F.L.)

**Keywords:** potato late blight, pseudomonads, hydrogen cyanide, biocontrol

## Abstract

Plants face many biotic and abiotic challenges in nature; one of them is attack by disease-causing microbes. *Phytophthora infestans*, the causal agent of late blight is one of the most prominent pathogens of the potato responsible for multi-billion-dollar losses every year. We have previously reported that potato-associated *Pseudomonas* strains inhibited *P. infestans* at various developmental stages. A comparative genomics approach identified several factors putatively involved in this anti-oomycete activity, among which was the production of hydrogen cyanide (HCN). Here, we report the relative contribution of HCN emission to the overall anti-*Phytophthora* activity of two cyanogenic *Pseudomonas* strains, *P. putida* R32 and *P. chlororaphis* R47. To quantify this contribution, we generated HCN-negative mutants (Δhcn) and compared their activities to those of their respective wild types in different experiments assessing *P. infestans* mycelial growth, zoospore germination, and infection of potato leaf disks. Using in vitro experiments allowing only volatile-mediated interactions, we observed that HCN accounted for most of the mycelial growth inhibition (57% in R47 and 80% in R32). However, when allowing both volatile and diffusible compound-mediated interactions, HCN only accounted for 1% (R47) and 18% (R32) of mycelial growth inhibition. Likewise, both mutants inhibited zoospore germination in a similar way as their respective wild types. More importantly, leaf disk experiments showed that both wild-type and Δhcn strains of R47 and R32 were able to limit *P. infestans* infection to a similar extent. Our results suggest that while HCN is a major contributor to the in vitro volatile-mediated restriction of *P. infestans* mycelial growth, it does not play a major role in the inhibition of other disease-related features such as zoospore germination or infection of plant tissues.

## 1. Introduction

In nature, plants face a variety of biotic and abiotic challenges such as climate change and infection by disease-causing microorganisms. Late blight caused by *Phytophthora infestans* (a hemibiotrophic pathogen) is one of the most devastating diseases in potato [1]. Worldwide, *P. infestans* is associated with a cost of several billion dollars per year, including the control efforts and production losses [2]. The current control methods mostly rely on the use of synthetic fungicides or copper-based products, both of which represent environmental hazards [3,4]. As an alternative tool to control plant diseases, researchers have focused their interest on identifying biocontrol agents [5], many of which are isolated from the host plants or soil [6]. Biocontrol agents (BCAs) are a group of microorganisms which show potential to restrict plant diseases either by inducing plant defense responses and/or by directly inhibiting the pathogen by producing antimicrobial agents [7]. Among the antimicrobial agents, hydrogen cyanide (HCN) has been reported by many studies to be a key biocontrol trait and was shown to be produced by multiple *Pseudomonas* species [8,9,10,11,12,13]. It is generally considered as a secondary metabolite produced at the end of the exponential phase and the start of the stationary phase. It is synthesized by hcnABC, which oxidizes glycine to produce HCN and CO_2_ [14]. HCN is a strong inhibitor of cytochrome c oxidase and other metalloenzymes and hence is a toxic molecule for all aerobic organisms. Nevertheless, cyanogenesis has been reported in many plants, fungi, myriapods, beetles, butterflies, and bacteria [15,16]. Many organisms have developed different strategies to avoid cyanide intoxication such as chemical conversion of HCN to thiocyanate by the rhodanese enzyme or the use of cyanide-insensitive oxidases [17,18,19].

In a recent comparative genomics study investigating the putative genetic determinants underlying the anti-*Phytophthora* activity of potato-associated *Pseudomonas* strains, we observed that the strains showing strong in vitro inhibition of *P. infestans* mycelial growth all harbored the HCN locus in their genome [20]. We therefore asked the question of the relative contribution of this particular biocontrol trait to the overall protective effect of cyanogenic *Pseudomonas* known for their strong biocontrol activity against different pathogens like *Phytophthora infestans*, *Helminthosporium solani*, *Rhizoctonia solani*, *Fusarium oxysporum,* and *Dickeya dianthicola* [6,21]. We selected *P. chlororaphis* (R47) and *P. putida* (R32), which were shown to inhibit mycelium growth and zoospore release in vitro but also to restrict the infection by *P. infestans* in leaf disk experiments [22]. The main aim of this study was therefore to quantify the relative contribution of HCN in the overall anti-oomycete activity of two efficient biocontrol *Pseudomonas* strains by generating HCN-negative mutants and assessing their effect on the different life stages and leaf infection ability of *Phytophthora infestans.* Understanding the mechanisms underlying the efficient inhibition of different developmental stages of *P. infestans* by biocontrol agents could help us in designing better crop protection strategies against the devastating late blight disease.

## 2. Materials and Methods 

### 2.1. Strains and Culture Media

*Pseudomonas* strains R47 and R32 [6] were routinely grown on Luria-Bertani (LB) agar medium made with 20 g/L of Difco LB broth (Lennox, Fisher Bioreagents, Geel, Belgium) and 15 g/L of agar, (Agar-agar, Kobe I, Carl Roth, Karlsruhe, Germany). *Phytophthora infestans* (Rec01) originally obtained from Agroscope Reckenholz was grown on unclarified V8 medium for dual assays and routine culture. To prepare unclarified V8 agar medium, 100 mL/L V8 juice (V8 Vegetable Juice, Campbell Soup Company, Camden, NJ, USA), 1 g/L calcium carbonate (Fluka Chemika, Buchs, Switzerland), and 15 g/L agar were dissolved in 1 L of distilled water and autoclaved. To prepare clarified V8 medium, the V8 juice was centrifuged twice at 5000 rpm for 15 min, and then the supernatant was used as above.

### 2.2. Generation of Δhcn Strains

For the generation of HCN-negative mutant strains (Δhcn), an allelic replacement technique using the I-SceI system was used [10,23]. A 2.8kb deletion was created within the HCN locus (hcnABC locus–3 kb). A suicide plasmid pEMG with a kanamycin resistance cassette and restriction sites for I-SceI, EcoRI, and BamHI was used. Two 500bp fragments were PCR-amplified from both ends of the HCN locus retaining the start codon of hcnA and the stop codon of the hcnC gene. Briefly, fragments 1 & 2 were generated by PCR amplification from the genomic DNA (gDNA) of R47 using the primers hcn47-1, hcn47-2 and hcn47-3, hcn47-4 containing the EcoRI-KpnI restriction sites for fragment 1 and the KpnI-BamHI restriction sites for fragment 2 (Appendix A). Fragments and pEMG were digested with their respective restriction enzymes, and the digested products were then triple-ligated to generate pEMG::Δhcn_R47 plasmid (Appendix A). Similarly, pEMG::Δhcn_R32 plasmid was generated using the primers hcn32-1, hcn32-2 and hcn32-3, hcn32-4. The final pEMG::Δhcn_R47 and pEMG::Δhcn_R32 plasmids were then transformed into R47 wild-type and R32 wild-type strains, respectively, to allow the plasmid to integrate into the chromosome using homologous recombination (HR). Following the HR event, kanamycin-resistant colonies were selected and screened for the presence of the inserted plasmid in gDNA. Positive clones were then transformed with pSW-2 plasmid, which contained the cassette for gentamycin resistance and the gene for the inducible production of I-SceI enzyme. Then, gentamycin-resistant clones were induced with *m*-toluate for the production of I-SceI enzyme, which facilitated the second crossover event to excise the suicide plasmid pEMG and HCN locus from the chromosome. Clones were then screened for their sensitivity to kanamycin. Kanamycin-sensitive clones were screened for the presence of mutated HCN fragments by PCR amplification using primers hcn47-1 and hcn47-4 for R47 and hcn32-1 and hcn32-4 for R32. The putative mutant strains were confirmed by a cyanide detection assay (refer to Section 2.4) (Appendix A).

### 2.3. Growth Curves

Two to three bacterial colonies from a 1-day-old plate were inoculated in LB broth for 16 h and then centrifuged at 5000 rpm for 3 min. The pellet was washed twice with 0.9% NaCl and then adjusted to OD_600_ = 1. Clarified V8 liquid medium (95 μL) was inoculated with 5 μL of bacterial suspension and grown for 20 h at 30 °C with shaking in a 24-well plate (Costar, Corning, NY, USA). A Cytation5 plate reader (BioTek, Winooski, VT, USA) was used to read absorbance every 30 min at 600 nm. The experiment was replicated twice with 3 technical replicates each. Uninoculated clarified V8 liquid medium was used as blank. 

### 2.4. Cyanide Detection Assay

Filter papers were soaked in a premixed solution of 10 mL chloroform (Fisher Scientific, Leicestershire, UK), 50 mg copper(II) ethylacetoacetate (Sigma-Aldrich, St. Louis, MO, USA), and 50 mg 4,4-methylenebis(N,N-dimethylaniline) (Sigma Aldrich, St. Louis, MO, USA) and allowed to dry overnight at room temperature [24]. These filter papers were then placed in one empty compartment of split-plate Petri dishes, and the other compartment was filled with LB agar. The LB side of the Petri dish was then inoculated with 3 drops (10 μL each) of bacterial suspension, which was pre-adjusted to OD_600_ = 1. The images were taken after 24 h, and the experiment was repeated twice with 3 technical replicates each. Uninoculated LB medium was used as negative control. 

### 2.5. Dual Assays

For volatile compound-mediated dual assays, split Petri dishes were used and both sides were filled with V8 agar medium. For diffusible compound-mediated assays, full plates with V8 agar medium were used. For both types of experiment, *Pseudomonas* strains and *P. infestans* were inoculated on the same day. Three drops of 10 μL bacterial suspension (OD_600_ = 1) were inoculated, and a plug from 2-week-old plates of *P. infestans* was used. Plates were incubated at 20 °C in the dark (upside down) for 2 weeks for full plate assays and for 1 week for split plate assays. Images were taken after the experiment and used for mycelial growth analysis using the ImageJ software. The experiment was repeated thrice with 3–4 technical replicates each. Mycelial growth in control plates (with no bacterial inoculation) was considered as 100%, and relative growth percentages were calculated for the mycelial growth in inoculated samples. Inhibition percentage was calculated by subtracting mycelial growth percentage from 100.

### 2.6. Zoospore Germination Experiment

Zoospores were harvested from sporangia present in a 2-week-old *P. infestans* plate after treatment with ice-cold sterile water for 2 h at 4 °C followed by 20 min at room temperature. Zoospores were harvested by pipetting from the water surface and counted using Jessen cell counting chamber. The number of zoospores was adjusted to 150,000 zoospores per mL. Zoospores were mixed in 1:1 ratio with bacterial suspension (in 0.9% NaCl adjusted to OD_600_ = 1) and incubated for 3 h in the dark at room temperature. After incubation, 30 μL of the samples were pipetted into the wells of a 24-well plate (Costar, Corning, NY, USA), and 4 images were taken randomly at 4× magnification using a Cytation5 plate reader. The percentage of germination was calculated by counting the total number of zoospores and the number of germinated zoospores on the images. Inhibition percentage was calculated by subtracting zoospore germination percentage from 100. The experiment was repeated thrice with 3 technical replicates each. 

### 2.7. Leaf Disk Experiment

Leaf disks were obtained from 3–4 independent potato plants (aged ~4 weeks, Bintje cultivar) and placed on 0.8% water agar medium in Petri dishes. Zoospores were harvested as mentioned in the zoospore germination experiment. The bacterial suspensions in 0.9% NaCl adjusted to OD_600_ = 1 were mixed with zoospores in a 1:1 ratio, and 10 μL of the mix was spotted on the abaxial surface of each leaf disk. For this experiment, plates were incubated in a humid chamber (polystyrene box with wet papers on the bottom) for 6 days at 18 °C in the dark. Pictures were taken after 6 days of inoculation with zoospores. Later, leaf disks were harvested and stored in 0.05% BHT in MeOH at −80 °C until fatty acid methyl esters (FAMEs) analysis. The experiment was repeated twice with 12–16 leaf disks from 3–4 plants in each experiment.

### 2.8. Fatty Acid Methyl Esters Analysis (FAMEs)

For this assay, leaf disks were harvested and processed using acid-catalysed transesterification to obtain FAMEs. Three leaf disks were pooled and treated with 1 mL of 5% H2SO4 in MeOH (*v/v*) and 50 μL of 0.05% butylated hydroxytoluene (*w/v*) in MeOH. Ten micrograms of glyceryl triheptadecanoate (Sigma-Aldrich, St. Louis, MO, USA) were used as an internal standard. Samples were placed in 7 mL glass tubes and were incubated at 85 °C for 45 min followed by cooling of tubes at room temperature and addition of 1.5 mL of 0.9% NaCl (*w/v*) and 2 mL of n-hexane. Samples were mixed by shaking for 5 min and centrifuged at 240 g for 5 min to separate the organic phase. The organic phase was then transferred into a new 7 mL glass tube, and the extraction was repeated two additional times with 2 mL of *n*-hexane. The pooled organic phases containing FAMEs were evaporated under a stream of nitrogen gas, and the FAMEs were resuspended in 200 μL heptane. FAMEs (2 μL) were injected into a gas chromatograph coupled to a flame ionization dectector (GC-FID) in split mode (50:1) for separation and quantification. The capillary column used was DB-23 by Agilent technologies (30 m × 250 μm × 0.25 μm). Supelco 37 component FAME mix from Sigma was used as a standard. The identification of Eicosapentaenoic acid was performed as described [25]. 

## 3. Results

### 3.1. Contribution of HCN to the Inhibition of P. infestans Mycelium Growth

Cyanide-negative mutants were generated in *Pseudomonas putida* R32 and in *Pseudomonas chlororaphis* R47 strains. A qualitative detection assay revealed the expected absence of HCN emission in both mutants (Appendix A). We performed two types of dual assays to quantify the impact of HCN on the mycelium growth of *P. infestans*. First, a volatile compound-mediated dual assay was performed to quantify the relative contribution of HCN on the overall volatile-mediated activity of R47 and R32. Second, a diffusible compound-mediated dual assay was performed to include the effects of non-volatile compounds (e.g., siderophores, phenazine) in the overall activity of R47 and R32 against *P. infestans* mycelial growth. In volatile compound-mediated assays, R47 wt reduced mycelium growth by 90% while R47 ∆hcn strain only reduced it by 33% (Figure 1a). Similarly, when exposed to the volatiles of R32 wt, mycelium growth was reduced by 84% but only by 4% when exposed to the volatiles from R32 ∆hcn (Figure 1b). Diffusible compound-mediated assay showed that the lack of HCN emission did not significantly impact the inhibition potential of R47 against *P. infestans* mycelial growth. It showed that R47 wt restricted the mycelial growth by 99%, while R47 ∆hcn restricted it by 98% (Figure 1c). Similarly, R32 wt was able to inhibit the mycelial growth by 98%, while an inhibition of 80% was observed in co-cultivation with R32 ∆hcn strain (Figure 1d). We also observed that both volatiles and diffusible compounds produced by R32 ∆hcn altered the mycelium phenotype as visualised by thin peripheral mycelium density in the volatile compound-mediated assay and the grainy mycelium phenotype in diffusible compound-mediated assay (Figure 1b,d).

### 3.2. Contribution of HCN to the Inhibition of P. infestans Zoospore Germination

Exposing harvested zoospores to cyanogenic wild types and their HCN-negative mutants revealed that both mutants retained the ability to significantly inhibit the germination of zoospores as compared to the control. The inhibition percentage upon exposure to R47 decreased from 88% for wt to 71% for the mutant, while the ∆hcn mutant strain of R32 caused even stronger inhibition of zoospore germination compared to its wt counterpart. Exposure to R32 wt led to 70% inhibition of zoospores germination while exposure to R32 ∆hcn increased the inhibition percentage to 90% (Figure 2a). Representative images for the zoospores germination experiment are shown in Figure 2b.

### 3.3. Contribution of HCN to the Inhibition of Infection by P. infestans on Potato Leaf Disks

Following the in vitro experiments which showed that HCN only accounts for a small portion of the anti-oomycete activity of Pseudomonas strains in direct contact-mediated assays (diffusible compound-mediated assay and zoospore germination experiment), we performed leaf disk experiments to quantify its relative contribution in the strains’ anti-*Phytophthora* activity in planta. Visual inspection of the leaf disks revealed that both R47 (wt and ∆hcn) and R32 (wt and ∆hcn) strains led to comparable levels of *P. infestans* infection on leaf disks, which were significantly lower than in the non-inoculated controls (Figure 3a). To quantify the infection level of *P. infestans* in plant tissues, we performed fatty acid methyl esters analysis of a *P. infestans*-specific fatty acid, the eicosapentaenoic acid (EPA; C20:5) [25]. Control leaf disks contained an average of 72 μg of EPA per sample, while leaf disks inoculated with either R47 wt, R47 ∆hcn, R32 wt, or R32 ∆hcn showed an average of 15, 18, 24, and 10 μg of EPA per sample, respectively (Figure 3b). No phytotoxic effect was observed when the bacterial strains were inoculated alone (without *P. infestans*) on the leaf disks (Appendix A). 

## 4. Discussion

Even after more than 170 years since the most infamous plant epidemic in Europe, the Irish potato famine, its causative agent *P. infestans* continues to challenge us with severe outbreaks overcoming our attempts to control the disease [26,27]. *Pseudomonas* bacteria, which are known for their widespread distribution across habitats, their catabolic versatility, and their abilities to efficiently colonize roots as well as to produce a variety of anti-microbial compounds, present promising possibilities as biocontrol agents [28]. Morrison and colleagues showed in 2017 that *P. fluorescens* LBUM636 could restrict *P. infestans* infection of potato tubers [29]. Similarly, promising results were obtained in our previous studies exploring the potential of *Pseudomonas* strains isolated from the potato microbiome for late blight control [6,21,22,30,31]. We have recently identified some candidate genes putatively involved in this anti-*Phytophthora* activity, among which those encoding production of the respiratory toxin HCN [20]. Here, we verified the involvement of this known particular biocontrol trait and quantified its relative contribution to the overall anti-oomycete activity of two cyanogenic *Pseudomonas* using in vitro and leaf disk experiments. In our volatile-mediated dual assays, R32 wild type restricted the mycelial growth by 84% and R47 wild type by 90% compared to the unexposed *P. infestans* control. In contrast, R32 ∆hcn showed only 4% growth inhibition while R47 ∆hcn restricted the *P. infestans* mycelial growth by 33% (Figure 1a,b). This suggests that HCN might be the sole anti-oomycete volatile emitted by R32 on V8 medium, or that other active compounds might be produced in too low concentrations on this medium to lead to observable inhibition. In the case of R47, our result suggests that HCN is not the only volatile compound active against *P. infestans* but that it plays a major role. Although we did not investigate the entire volatilome of R47 grown on V8, we might speculate that the remaining inhibition activity could be due to volatile compounds such as 1-undecene, dimethyl disulfide, dimethyl trisulfide, S-methyl methanethiosulfonate, acetophenone, or 2-hexanone, which were previously shown to be produced by R47 and to have anti-*Phytophthora* activities [6,25,30]. When diffusible compounds were also allowed to participate in the *P. infestans*–*Pseudomonas* warfare, experiments showed no loss of activity of R47 ∆hcn compared to its wild ype (98 vs. 99% inhibition respectively, Figure 1c), and only moderate loss of activity (from 98 to 80%) for the mutant of R32 (Figure 1d). This finding reveals the presence of other determinants of mycelial growth inhibition in both strains, and of very strong one(s) in the case of R47. Based on the strains’ genome content [20], possible candidate molecules explaining the remaining inhibitory activity of the cyanide deficient mutants are e.g., pyoverdine, which is encoded in the genome of both strains, and phenazine, which is encoded only in R47 [20]. Phenazines have been previously shown to inhibit *P. infestans* infection on potato tubers [29], and the siderophore pyoverdine could also play an important role, as competition for iron is a well-known mechanism by which bacteria inhibit the growth of fungal pathogens [32]. 

Beyond mycelia, zoospores are an important life form of *P. infestans* that is of high relevance for infection. Our results indicated that more efficient compounds than HCN were likely involved in this process. Interestingly, the cyanide-deficient mutant inhibited zoospore germination more efficiently than the wild type (Figure 2a). This could have been caused by better growth due to the absence of the fitness cost generated by HCN biosynthesis, but the growth curves generated indicated that the deletion of hcnABC genes did not provide the mutant strains with growth advantages on V8 medium (Appendix A). Our current hypothesis is that independently of growth rate, strains lacking HCN might have compensated for the loss of this important weapon by increasing the production of other compounds, leading to slightly but significantly higher activity than the wild type in the *P. putida* R32 strain. Such a phenomenon could also explain the altered mycelium phenotype of *P. infestans* exposed to the R32 cyanide deficient mutant observed in both our dual assays (Figure 1b,d). This putative effect, which we are currently investigating, was however not observed in *P. chlororaphis* R47, where the cyanide-deficient mutant exhibited slightly but significantly lower inhibiting activity than the wild type, suggesting minor contribution of HCN to the inhibition of zoospore germination (Figure 2a).

Leaf disk infection assays showed that both ∆hcn strains reduced *P. infestans* infection on plant tissues to a similar extent as their corresponding wild types (Figure 3). A similar trend to higher efficacy of the mutant of *P. putida* R32 compared to the wild type could be observed, although this was not significant (Figure 3b). Since zoospores were used to infect leaf tissues, it might indicate that the mode of action of these bacterial strains lies in direct inhibition of the pathogen rather than in inducing defense-related genes in the plant host. 

In conclusion, our experiments show that HCN is the major (in R47) or only (in R32) contributor to the inhibition of *P. infestans* mycelial growth in volatile mediated experiments but that it is not involved in the inhibition of *P. infestans* by the biocontrol strains when they are in direct contact with the pathogen. However, this does not mean that the ability to emit HCN is not an important biocontrol trait in other experimental setups or in other types of plant–pathogen interactions. More importantly, our results highlight the occurrence of other, yet-unknown efficient compounds of great interest in the fight against *P. infestans* in the relatively small genome of *P. putida* R32 (ca 5.6 MB), which we aim to identify in the near future using a mutant bank approach. Moreover, the indication of higher activity or altered phenotype observed in the cyanide deficient mutant of this strain suggests that the presence of cyanogenic strains e.g., in synthetic communities designed as biocontrol consortia, might alter the metabolic potential of such consortia in a previously unsuspected way, an aspect that also deserves further investigation. 

## Figures and Tables

**Figure 1 microorganisms-08-01144-f001:**
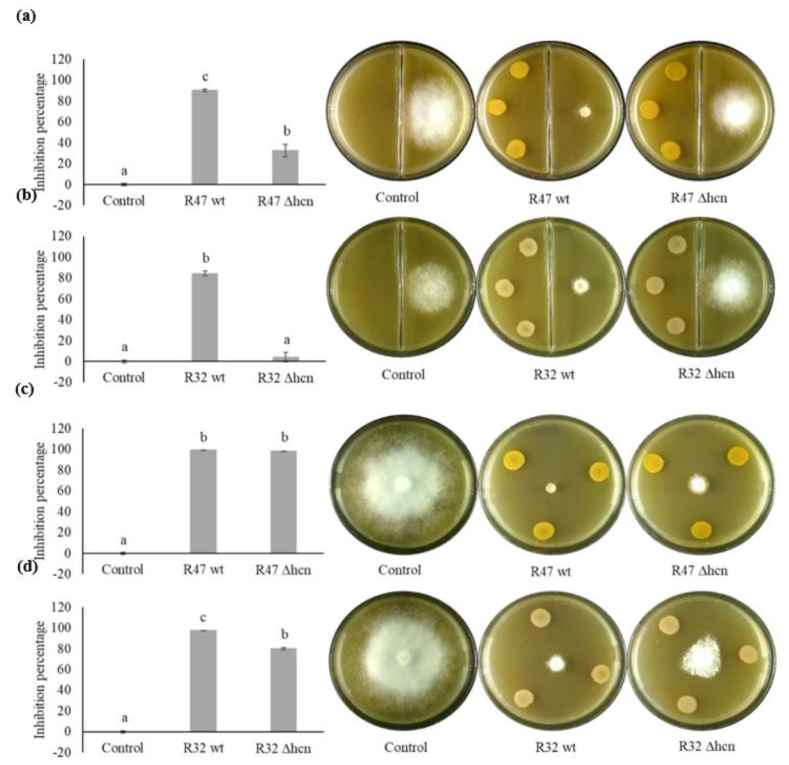
Effect of R47 (wt and ∆hcn) and R32 (wt and ∆hcn) strains on *Phytophthora infestans* mycelium growth. (**a**,**b**) Volatile compound-mediated dual assay; (**c**,**d**) diffusible compound-mediated dual assay. Left panels: quantification of mycelium growth of *P. infestans* represented as inhibition percentages. The bars show averages (*n* = 3 independent assays with 3–4 technical replicates) with error bars indicating standard error of the mean. Letters indicate significant differences between treatments as per ANOVA followed by Tukey’s honestly significant difference (HSD) (threshold *p* < 0.05); Right panels: representative images of dual assays corresponding to the graphs on the left.

**Figure 2 microorganisms-08-01144-f002:**
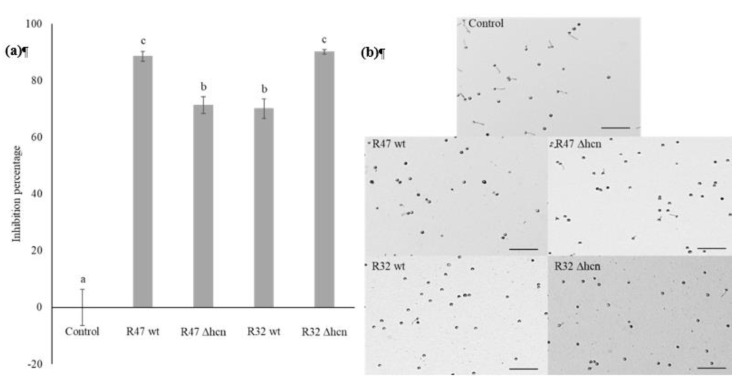
Effect of R47 (wt and ∆hcn) and R32 (wt and ∆hcn) strains on zoospore germination. (**a**) Graph representing the inhibition percentages of germinated zoospores when exposed to different strains and control (unexposed to bacterial strains). The bars show averages (n = 3 independent assays with 3–4 technical replicates) with error bars indicating standard error of the mean. Letters indicate significant differences between treatments as per ANOVA followed by Tukey’s HSD (threshold *p* < 0.05). (**b**) Representative images of zoospores after 3 h incubation with or without bacterial strains at room temperature. Scale bar = 100 μm.

**Figure 3 microorganisms-08-01144-f003:**
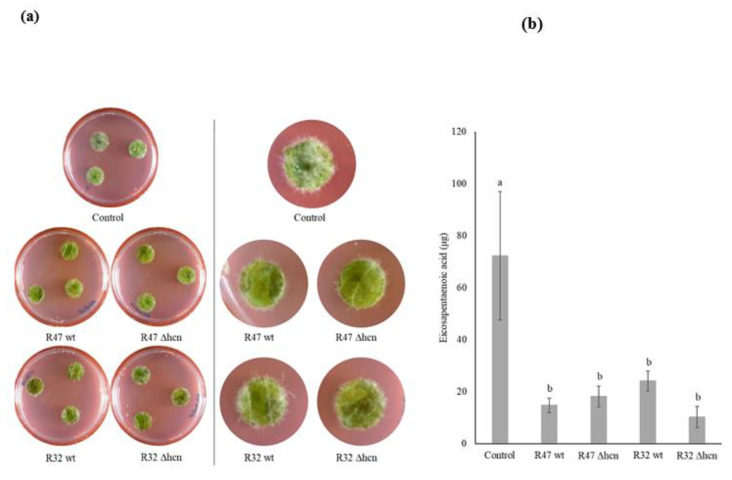
*Pseudomonas* strains (R32 and R47) restrict the infection by *P. infestans* on leaf disks. (**a**) Left panel: representative images of leaf disks co-inoculated with *Pseudomonas* strains and *P. infestans* zoospores. Pictures were taken 6 days post-inoculation. Right panel: zoomed images of leaf disks showing *P. infestans* infection (white hyphal mat). (**b**) Quantification of infection by measuring *P. infestans*-specific fatty acid (EPA; C20:5) content in leaf tissues. The bars show averages (*n* = 3) with error bars indicating standard error of the mean. Letters indicate significant differences between treatments as per ANOVA followed by Tukey’s HSD (threshold *p* < 0.05).

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
