# Peer review of "Contribution of Hydrogen Cyanide to the Antagonistic Activity of Pseudomonas Strains Against Phytophthora infestans"

_microorganisms, 2020, doi:10.3390/microorganisms8081144_

Round 1

Reviewer 1 Report

In the manuscript the authors presented the results of experiments aimed to understand the contribution of hydrogen cyanide to the antagonistic activity of Pseudomonas strains against Phytophthora infestans. The methods described are trustable and repeatable and the results are well presented. Based on the results showed, they finally conclude that even if HCN is the major or only contributor to the inhibition of P. infestans mycelial growth in a volatile experiments, it is not involved directly in the inhibition of the pathogen. This opens to further work aimed to study the complex and fascinating interaction among the three main actors: plant- pathogen- biocontrol agent. All in all, I recommend the acceptance of the manuscript for publication.

Author Response

Thank you for your positive assessment of our paper!

Reviewer 2 Report

The manuscript of Anand et al. presents the results of the study on biocontrol effect of Pseudomonas strains against Phytophthora infestans, the causal agent of late blight. In their study, the authors assessed the contribution of HCN emission to anti-Phytophthora activity with use of hcn negative mutants. As a result, it was shown that HCN was a major contributor to the in vitro volatile-mediated restriction of P. infestans mycelial growth but it didn’t play a major role in the inhibition of other disease-related features such as zoospore germination or infection of plant tissues.

The results obtained in the study are significant and provide an advance in current knowledge. The quality of presentation is high, and the study is technically sound. Thus, the manuscript can be accepted for publication in the present form.

I have a minor question regarding the hcn locus. Can the hcn locus be involved in other processes, in addition to the HCN biosynthesis, and, therefore, can the deletion of the locus have other effects? The relevant information can be added to the Introduction section.

Also please check the use of italics in the latin names in the Results section (e.g., line 167-168, 170, 174, 216-220).

Author Response

Thank you very much for your positive assessment of our paper and careful review. We have changed the font of the latin names (see track change version). Regarding your question on a putative different role of the hcn locus, we do not think that there is such a different and direct role of the locus, especially since the higher zoospore-inhibiting activity of the mutant was only observed in one of the two strains. However, we think that HCN itself (the molecule) might have effects on the producing cells (e.g. by repressing or inducing expression of other biocontrol genes), and not only on the target organisms (P. infestans). We however did not add this hypothesis to the introduction (as suggested), because we think this needs further investigation and validation.